# Optimal Transport under Group Fairness Constraints

## Abstract

Ensuring fairness in matching algorithms is a key challenge in allocating scarce
resources and positions. Focusing on Optimal Transport (OT), we introduce a novel
notion of group fairness requiring that the probability of matching two entities from
any two given groups in the OT plan follows a user-specified target. We develop two
relaxation strategies to solve this constrained problem. The first one involves solv-
ing a penalized OT problem, for which we derive novel finite-sample complexity
guarantees. Our second strategy leverages bi-level optimization to learn a ground
cost that produces a fair OT solution, which can be reused to match new samples.

## 1 Introduction

Algorithmic matching mechanisms play an increasing role in modern societies, handling the distri-
bution of rare goods by connecting individuals or firms through algorithmic decisions rather than
through price driven markets. Examples of such mechanisms include online job recommendations,
college admissions systems, and kidney allocation circuits. An increasing concern is the fairness
of such mechanisms. For instance, school assignment systems have been criticized for excessively
matching students from privileged backgrounds with elite institutions regardless of their academic
potential [25, 67]. These concerns stem from the fact that the matching decision made by the central
algorithm is not independent of a group-defining attribute such as social background. Although group
fairness has been extensively explored in supervised and unsupervised learning settings [4], extending
these notions to matching problems requires novel formal definitions and algorithmic tools.

In this work, we study group fairness in matching through optimal transport (OT), which has been a
long-standing tool to model matching problems in economics and social sciences since the pioneering
work of Kantorovitch [32, 48, 9, 54, 63, 39]—see Appendix A for a detailed survey of related work.
Our contribution is twofold. Our first contribution is a formal framework for fair optimal transport
under group fairness constraints, featuring two new fairness definitions designed for this setting.
Secondly, we design two algorithms that rely on penalized optimal transport and bi-level optimization,
respectively, to achieve group fairness in optimal transport. For the first approach, we derive a novel
sample complexity bound, which could be of independent interest for general penalized entropic
optimal transport problems. All proofs are given in the Appendix.

## 2 Formalizing Group Fairness in Optimal Transport

### 2.1 Optimal Transport

**Entropic Optimal Transport.** We consider the entropic regularized optimal transport problem
[61, 34, 27, 62, 44] between two probability distributions $\mu$ and $\eta$ with ground cost $c$:

$$\mathbf{W}_c^\varepsilon(\mu, \eta) := \min_{\pi \in \Pi(\mu, \eta)} \int_{\mathbb{R}^d \times \mathbb{R}^d} c(x, y) \, d\pi(x, y) + \varepsilon \mathbf{KL}\big(\pi | \mu \otimes \eta\big) \ ,$$

Submitted to 39th Conference on Neural Information Processing Systems (NeurIPS 2025). Do not distribute.

where the minimum is taken over couplings $\Pi(\mu, \eta)$ of $\mu$ and $\eta$ and where $\mathbf{KL}$ denotes the Kullback-Leibler (KL) divergence (see [19]). If $\mu$ and $\nu$ are sums of Dirac measures, *i.e.*, if $\mu = 1/n \sum_{i=1}^{n} \delta_{\mathbf{x}_i}$ and $\eta = 1/m \sum_{j=1}^{m} \delta_{\mathbf{y}_j}$, the problem above reduces to

$$\mathbf{W}_{\mathbf{C}}^{\varepsilon} := \min_{\mathbf{\Pi} \in \Pi} \sum_{i=1}^{n} \sum_{j=1}^{m} \mathbf{\Pi}_{ij} \mathbf{C}_{ij} + \varepsilon \mathbf{KL}(\mathbf{\Pi}) \ ,$$

where $\Pi := \{\mathbf{\Pi} \in \mathbb{R}_+^{n \times m} \mid \mathbf{\Pi} \mathbf{1}_m = 1/n \mathbf{1}_n, \mathbf{\Pi}^\top \mathbf{1}_n = 1/m \mathbf{1}_m\}$, $\mathbf{KL}(\mathbf{\Pi}) := \sum_{ij} \mathbf{\Pi}_{ij} \log \mathbf{\Pi}_{ij}$ and $\mathbf{C} \in \mathbb{R}^{n \times m}$ with $\mathbf{C}_{ij} := c(x_i, y_j)$. The KL divergence is strictly convex, and consequently, we denote the corresponding unique minimizer by $\mathbf{\Pi}_{\varepsilon}(\mathbf{C})$.

## 2.2 Fair Optimal Transport

We consider two distributions $\mu \in \mathcal{P}(\mathcal{X} \times \mathcal{S})$ and $\eta \in \mathcal{P}(\mathcal{Y} \times \mathcal{W})$ to be matched, where $\mathcal{X}$ and $\mathcal{Y}$ represent feature spaces while $\mathcal{S}$ and $\mathcal{W}$ correspond to sensitive attributes (e.g., gender, ethnicity or age) defining groups of entities in $\mu$ and $\eta$ respectively. We restrict ourselves to the case where $\mathcal{X}$ and $\mathcal{Y}$ are compact subspaces of $\mathbb{R}^k$ for the rest of this paper. Additionally, $\mathcal{W}$ and $\mathcal{S}$ are assumed to be finite sets which we identify, respectively, with $\{1, \ldots, K_w\}$ and $\{1, \ldots, K_s\}$. We let $p_i = \mathbb{P}(S = i)$ and $q_j = \mathbb{P}(W = j)$ the probability of each group $i$ in $\mu$ and $j$ in $\eta$, respectively, and denote by $\mathbf{p}$ and $\mathbf{q}$ the resulting distributions over $\mathcal{W}$ and $\mathcal{S}$.

**Fairness targets.** Optimal transport imposes structure on the coupling of two probability distributions through the cost function, making it more likely to match "nearby" points.

**Example 2.1.** *Consider a school assignment system, in which students are characterized by their geographic position $X \in \mathbb{R}^2$ and a social status $S \in \{\text{high}, \text{low}\}$. Similarly, schools are characterized by geographic positions $Y \in \mathbb{R}^2$ and a prestige level $W \in \{\text{elite}, \text{non-elite}\}$. Suppose that students with $S = \text{high}$ are more likely to live near elite schools, which are often situated in privileged neighborhoods, while students with low social status tend to live near non-elite schools. In this setting, assigning students to schools using optimal transport with Euclidean cost—minimizing the distance between students and schools—will result in highly segregated schools.*

In this example, the block-sparse structure of the optimal transport plan can be seen as a source of unfairness, as the matching will be highly correlated with the social status of the students and the elitist nature of the schools. To define a fair transport plan, we assume that we are given a *fairness target* $\mathbf{F}$, which is a $K_s \times K_w$ matrix that specifies, for each pair of groups $(s, w) \in \mathcal{S} \times \mathcal{W}$, the desired probability of matching members of group $s$ with group $w$. To be valid, the matrix $\mathbf{F}$ should itself be a coupling between $\mathbf{p}$ and $\mathbf{q}$: it should be non-negative and satisfy the constraints

$$\sum_{w=1}^{K_w} \mathbf{F}_{sw} = p_s \text{ and } \sum_{s=1}^{K_s} \mathbf{F}_{sw} = q_w$$

for all $(s, w)$. We hence write $\Pi(\mathbf{p}, \mathbf{q})$ the set of fairness targets.

**Example 2.2.** *Consider the segregated schooling system introduced in Example 2.1. Let $p := \mathbb{P}(S = \text{low})$ and $q := \mathbb{P}(W = \text{non-elite})$. A newly appointed city administrator wishes to strongly limit social homogamy within public schools, and hence requires $60\%$ of student with low social status to be matched to elite schools. The fairness target can be written as*

$$\mathbf{F} = \begin{bmatrix} 0.4 \times p & 0.6 \times p \\ q - 0.4 \times p & 1 - q - 0.6 \times p \end{bmatrix}. \tag{1}$$

**Cost-Insensitive Fairness.** We can now define a notion of *fair coupling* with respect to a given fairness target $\mathbf{F}$.

**Definition 2.3** (**F**-Fair coupling). *Let $\mu \in \mathcal{P}(\mathcal{X} \times \mathcal{S})$ and $\nu \in \mathcal{P}(\mathcal{Y} \times \mathcal{W})$. A coupling $\pi \in \Pi(\mu, \nu)$ is said to be **F**-**fair** if for all $(s, w) \in \mathcal{S} \times \mathcal{W}$ we have*

$$\pi_{SW}(s, w) = \mathbf{F}_{sw} \ ,$$

*where $\pi_{SW}$ is the coupling on $\mathcal{S} \times \mathcal{W}$ that is obtained from $\pi$ by marginalizing over $x$ and $y$, that is*

$$\pi_{SW}(s, w) := \pi(\mathcal{X} \times \{s\} \times \mathcal{Y} \times \{w\}) = \int_{\mathcal{X} \times \{s\} \times \mathcal{Y} \times \{w\}} d\pi(x, u, y, t) \ .$$

In other words, a coupling is fair if the amount of mass moved from group $s$ to group $w$ is equal to $\mathbf{F}_{sw}$. We denote by $\Pi_{\text{fair}}^{\mathbf{F}}(\mu, \eta)$ the set of $\mathbf{F}$-fair couplings and define the corresponding fair optimal transport problem as the problem of finding

$$\Pi_{\varepsilon,\text{fair}}^{\star} \in \operatorname*{arg\,min}_{\pi \in \Pi_{\text{fair}}^{\mathbf{F}}(\mu, \eta)} \int c(x,y)\, d\pi(x,y) + \varepsilon \mathbf{KL}(\pi | \mu \otimes \eta) \ . \tag{2}$$

**Proposition 2.4.** *Assume that $\mu$ and $\eta$ have compact support. Then, for any $\mathbf{F} \in \Pi(\mathbf{p}, \mathbf{q})$, there exists a unique fair optimal transport plan.*

**Cost-Sensitive Fairness.** In some scenarios, it is desirable to enforce fairness while taking the ground cost $c$ into account. Instead of requiring that a fraction $\mathbf{F}_{sw}$ of all matches occur between individuals from groups $s$ and $w$, we may require that these groups collectively bear a share $\mathbf{F}_{sw}$ of the total transport cost. This leads to the following definition.

**Definition 2.5** (Cost Fair OT). *Let $\mu \in \mathcal{P}(\mathcal{X} \times \mathcal{S})$ and $\nu \in \mathcal{P}(\mathcal{Y} \times \mathcal{W})$. A coupling $\pi \in \Pi(\mu, \nu)$ is said to be **F-cost fair** if for for all $(s, w) \in \mathcal{S} \times \mathcal{W}$ we have*

$$\int_{\mathcal{X} \times \{s\} \times \mathcal{Y} \times \{w\}} c(x,y)\, d\pi(x,u,y,t) = \mathbf{F}_{sw}\bar{c} \ ,$$

*where $\bar{c} := \mathbb{E}_{\mu \otimes \eta}\big[c(X,Y)\big]$ is the total cost.*

**Example 2.6.** *Consider again the segregated school system introduced in previous examples. Enforcing cost-sensitive fairness in this context involves weighting matches according to the distance between students and schools, thereby discouraging matches that are overly costly.*

**Finite-Sample Fair Optimal Transport.** Consider two datasets $(\mathbf{x}_i, \mathbf{s}_i)_{i=1}^{n} \in \mathcal{X} \times \mathcal{S}$ and $(\mathbf{y}_i, \mathbf{w}_i)_{i=1}^{m} \in \mathcal{Y} \times \mathcal{W}$ drawn i.i.d. from $\mu$ and $\nu$ respectively. The finite sample version of the fair optimal transport (2) corresponds to replacing $\mu$ and $\nu$ by their empirical counterparts, that is, $\mu_n = 1/n \sum_i \delta_{(x_i,s_i)}$ and $\nu_n = 1/m \sum_j \delta_{(y_j,w_j)}$. Let $\mathbf{B}_{sw} := \big(\mathbb{1}_{\mathbf{s}_i = s} \mathbb{1}_{\mathbf{w}_j = w}\big)_{i,j} \in \{0,1\}^{n \times m}$ and the sample problem is thus given by

$$\min_{\mathbf{\Pi} \in \Pi} \operatorname{Tr}\big[\mathbf{\Pi}^{\top} \mathbf{C}\big] + \varepsilon \mathbf{KL}\big(\mathbf{\Pi}\big) \tag{3}$$

$$\text{s.t. } \forall (s, w) \in \mathcal{S} \times \mathcal{W},\ \operatorname{Tr}\big[\mathbf{\Pi}^{\top} \mathbf{B}_{sw}\big] = \sum_{i | \mathbf{s}_i = s} \sum_{j | \mathbf{w}_i = w} \mathbf{\Pi}_{ij} = \mathbf{F}_{sw} \ . \tag{4}$$

For cost-sensitive fairness, (4) is replaced by $\operatorname{Tr}\big[\mathbf{B}_{sw} \mathbf{\Pi}^{\top} \mathbf{C}\big] = \bar{c} \mathbf{F}_{sw}$ with $\bar{c} := (nm)^{-1} \sum \mathbf{C}_{ij}$. This problem can be solved using a modified Sinkhorn algorithm—see Appendix B for details.

# 3  Two Strategies for Fair OT

**Relaxing Fairness Constraints.** Enforcing exact fairness can lead to transport plans with prohibitively high cost. In practice, it is often preferable to allow some tolerance to balance the trade-off between fairness and overall efficiency. In our setting, we aim to find a transport plan that is fair while remaining reasonably close to the original plan. To achieve this, we adopt a $\rho$-relaxed fairness approach, requiring that for all $(s, w) \in \mathcal{S} \times \mathcal{W}$, the fairness violation satisfies $\mathcal{L}_{\mathbf{F}}(\mathbf{\Pi}) \leq \rho$, where

$$\mathcal{L}_{\mathbf{F}}(\mathbf{\Pi}) := \sum_{(s,w) \in \mathcal{S} \times \mathcal{W}} \left( \operatorname{Tr}\big[\mathbf{\Pi}^{\top} \mathbf{B}_{sw}\big] - \mathbf{F}_{sw} \right)^2, \qquad (\rho\text{-relaxed cost-insensitive fairness}) \tag{5}$$

or

$$\mathcal{L}_{\mathbf{F}}(\mathbf{\Pi}) := \sum_{(s,w) \in \mathcal{S} \times \mathcal{W}} \left( \operatorname{Tr}\big[\mathbf{B}_{sw} \mathbf{\Pi}^{\top} \mathbf{C}\big] - \bar{c} \mathbf{F}_{sw} \right)^2, \qquad (\rho\text{-relaxed cost-sensitive fairness}) \tag{6}$$

and $\rho \geq 0$ is a tolerance level that leads to a relaxation of (3)-(4) where (4) is replaced by $\mathcal{L}_{\mathbf{F}}(\mathbf{\Pi}) \leq \rho$. To find optimal transport plans under relaxed fairness constraints, we propose two strategies.

## 3.1 Fairness-Penalized OT

Our first strategy is a direct penalization strategy. We solve a penalized version of optimal transport problem under $\rho$-relaxed fairness by introducing a Lagrange multiplier associated to the constraint $\mathcal{L}_{\mathbf{F}}(\mathbf{\Pi}) \leq \rho$:

$$\mathbf{OT}_{\mathbf{F}}^{\varepsilon} := \min_{\mathbf{\Pi} \in \Pi} \mathrm{Tr}\big[\mathbf{\Pi}^{\top}\mathbf{C}\big] + \varepsilon\mathbf{KL}\big(\mathbf{\Pi}\big) + \lambda\mathcal{L}_{\mathbf{F}}(\mathbf{\Pi}) \ , \tag{7}$$

where $\lambda, \epsilon > 0$ control regularization strength and $\mathcal{L}_{\mathbf{F}}(\mathbf{\Pi})$ is one of the two fairness penalties introduced in Equations (5)-(6), which may depend on the cost. This is an instance of penalized entropic optimal transport which can be solved efficiently using a generalized conditional gradient algorithm [64].

**Sample Complexity of Penalized OT.** One of the main contributions of our work is to establish a sample complexity bound for the fairness penalized entropic optimal transport cost, by building upon results from entropic optimal transport [34, 55]. To formalize this, note that the optimization problem in (7) can be defined for arbitrary measures $\alpha \in \mathcal{P}(\mathcal{X} \times 0, 1)$ and $\beta \in \mathcal{P}(\mathcal{Y} \times 0, 1)$ as (see Appendix D.2 for a formal definition)

$$m^{\star}(\alpha, \beta) := \min_{\pi \in \Pi(\alpha, \beta)} \int_{\mathbb{R}^d \times \mathbb{R}^d} c(x, y) \, d\pi(x, y) + \varepsilon\mathbf{KL}\big(\pi \| \alpha \otimes \beta\big) + \lambda\mathcal{L}_{\mathbf{F}}(\pi) \ . \tag{8}$$

For simplicity, we restrict to the simple setting where which $K_s = K_w = 2$ and the fairness penalty is chosen to be cost-insensitive. We note, however, that our proof extends naturally to the more general case. Similar to [65] and [34], our proof relies on the fact that the measures are compactly supported. We also require the following assumption on the ground cost function.

**Assumption 1.** The cost function $c : \big(\mathcal{X} \times \{0, 1\}\big) \times \big(\mathcal{Y} \times \{0, 1\}\big) \to \mathbb{R}_+$ is Lipschitz and $\mathcal{C}^{\infty}$.

> **Theorem 3.1.** Assume for simplicity that $m = n$. Under Assumption 1, we have
> $$\mathbb{E}\big|m^{\star}(\mu_n, \eta_n) - m^{\star}(\mu, \eta)\big| \lesssim \frac{\log(n)}{\sqrt{n}} \ .$$

Comparing our result with existing sample complexity bounds for the *unpenalized* entropic OT problem [34, 55], we achieve similar bounds up to a logarithmic factor.

**Remark 1.** In the formulation above, the cost function is defined on inputs from $\mathcal{X} \times 0, 1$ and $\mathcal{Y} \times 0, 1$ for full generality. By setting the cost function $c$ to be independent of the sensitive attribute, we recover our original setting.

**Proof sketch.** Our proof works by finding adequate random variables $Y_n, Z_n$ such that

$$Y_n - m^{\star}(\mu, \eta) \leq m^{\star}(\mu_n, \eta_n) - m^{\star}(\mu, \eta) \leq Z_n - m^{\star}(\mu, \eta) \ . \tag{9}$$

*Lower bound.* We obtain $Y_n$ through linearization of the convex penalty, which is guaranteed to yield a proper lower bound through the connection between the optimality conditions of our penalized problem and its linearized counterpart [64]. This linearized problem may then be cast as an entropic optimal transport problem with modified cost, allowing us to leverage standard results on the sample complexity of entropic optimal transport [33, 55].

*Upper bound.* To obtain $Z_n$, we proceed in two steps. First, we evaluate the optimal population coupling $\pi_{\star}$ on the dataset, that is, we consider the random coupling

$$\widehat{\pi}_n^{\star} := \frac{1}{n^2} \sum_{ij} \frac{d\pi^{\star}}{d\mu \otimes \nu}((x_i, s_i), (y_j, w_j))\delta_{((x_i, s_i), (y_j, w_j))} \tag{10}$$

and let $\widehat{Z}_n := \int c \, d\widehat{\pi}_n^{\star} + \varepsilon\mathbf{KL}(\widehat{\pi}_n^{\star} \| \mu_n \otimes \nu_n) + \lambda\mathcal{L}_{\mathbf{F}}(\widehat{\pi}_n^{\star})$. Observe that $\hat{\pi}_n^{\star}$ does not necessarily satisfy the marginal constraints that define $\Pi(\mu_n, \eta_n)$, and, consequently, $\widehat{Z}_n$ is not necessarily an upper bound of $m_n^{\star}$. We thus secondly project $\widehat{\pi}_n^{\star}$ onto the constraint set. We use the `round`

algorithm proposed by Altschuler et al. [1, Algorithm 2] (see Appendix D.2.2). The rounded coupling $\bar{\pi}_n^\star = \text{round}(\widehat{\pi}_n^\star)$ now is a valid coupling of the empirical distributions, and we hence use

$$Z_n := \int c \, d\bar{\pi}_n^\star + \varepsilon \mathbf{KL}(\bar{\pi}_n^\star || \mu_n \otimes \nu_n) + \mathcal{L}_\mathbf{F}(\bar{\pi}_n^\star) \ . \tag{11}$$

We conclude the proof by using $|Z_n - m_\infty^\star| \le |Z_n - \widehat{Z}_n| + |\widehat{Z}_n - m^\star(\mu, \nu)|$. The first term can be bounded using regularity arguments and concentration of the projected coupling, and the second term using $\mathbb{E}[\widehat{Z}_n] = m^\star(\mu, \nu)$, independence of samples, and compactness arguments.

## 3.2 Bilevel Optimization Formulation

A second strategy builds on cost-learning in optimal transport [49] and consists in solving the bilevel optimization problem

$$\min_{\theta \in \Theta} \mathcal{L}_\mathbf{F}\big(\mathbf{\Pi}(c_\theta), \mathbf{C}, \Gamma\big) + \lambda^{-1} \mathscr{D}(c_\theta, c_\text{base}) \ \text{ s.t. } \ \mathbf{\Pi}(c_\theta) = \arg\min_{\mathbf{\Pi}} \text{Tr}\big[\mathbf{\Pi}^\top \mathbf{C}_\theta\big] + \varepsilon \mathbf{KL}\big(\mathbf{\Pi}\big) \tag{12}$$

where $\mathbf{C}_\theta := \big[c_\theta(\mathbf{x}_i, \mathbf{y}_j)\big]_{ij}$ denotes a parameterized family of cost matrices, $c_\text{base} : \mathcal{X} \times \mathcal{Y} \to \mathbb{R}+$ is a user-specified baseline cost function, and $\mathscr{D}$ is a discrepancy measure between cost functions that encourages the learned cost to remain close to the baseline. Problem (12) can be efficiently solved using gradient-based methods, as its inner component is an entropically regularized optimal transport problem that is strongly convex over the probability simplex. Typical approaches rely on iterative differentiation [7, 30, 53, 59] or (approximate) implicit differentiation [20, 21, 24, 35, 47, 60], and can accommodate a wide variety of parameterized costs. Relevant examples include Mahalanobis distances, $c_\mathbf{M}(x, y) = (x - y)^\top \mathbf{M}(x - y)$ where $\mathbf{M}$ is a PSD matrix, and their nonlinear counterpart, $c_\theta(x, y) = \|\phi_\theta(x) - \phi_\theta(y)\|_2^2$, where $\phi_\theta : \mathbb{R}^d \to \mathbb{R}^k$ is a neural network with parameters $\theta$.

Our bilevel cost-learning approach allows for high flexibility and modularity and yields an interpretable cost function, which can be readily reused on new samples. It however requires to solve an intricate non-convex problem and comes with no theoretical guarantees, in contrast to the penalized approach. We provide a thorough comparison of both approaches in Appendix C.

# 4 Numerical Illustrations

We illustrate our algorithms using simulated mixture of Gaussian data, focusing on cost-sensitive fairness. Figure 1 displays our results for the penalized approach. The left panel shows how the fairness loss evolves as the penalty increases. We observe that our matching gets increasingly fair with larger penalties. The center and right panels show the fairness loss as a function function of the cost difference and KL divergence, respectively, between the fair plan and the unfair plan (that is the plan computed with $\lambda = 0$). One notices that lower fairness loss corresponds to higher bias with respect to the unfair optimal transport plan. We conduct the same experiment with the cost learning approach. The corresponding results—reported in Appendix E—yield similar conclusions.

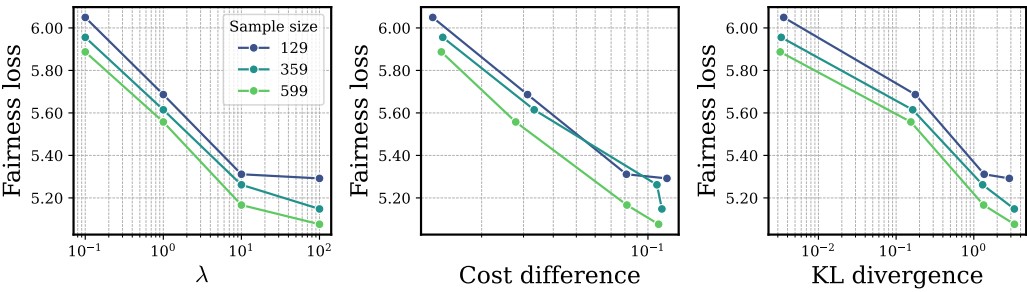

Figure 1: Fairness vs. penalty (**left**), fairness-cost difference (**center**) and fairness-KL divergence (**right**) trade-offs (w.r.t. the non-penalized problem) with varying sample sizes for the **penalized OT problem** with cost-sensitive fairness loss.

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

## A   Related Work

**Fairness in Supervised Learning.**   To model fairness issues, one usually considers in supervised learning datapoints of the form $(X, Y, S) \in \mathcal{X} \times \mathcal{Y} \times \mathcal{S}$, where $X$ represents an individuals' features, $Y$ is an outcome and $S$ is a sensitive attribute — such as gender, age or income. In its most common sens, statistical fairness is defined through independence between the sensitive attribute $S$ and the output of an algorithm $f$, that is an algorithm is fair if $f(X) = \widehat{Y} \perp\!\!\!\perp S$. To solve this problem, the literature on fair supervised learning has considered mutual information based penalties to enforce independence between the prediction and the sensitive attribute [3, 14, 52], and has also devised various criteria to be minimized as fairness objectives such as demographic parity (DP) [8, 23] or equality of odds (EO) [38].

**Fairness in Matching Mechanisms.**   Since the pioneering work of Gale and Shapley [31] on the stable marriage problem, a series of works have provided theoretical studies of the fairness of individual preference-based matching mechanisms [2, 43, 11, 22, 10, 12, 51, 70]. In the same time, a growing body of work has provided empirical assessment of fairness in matching mechanisms [40, 26, 68, 13, 18, 45, 42].

**Fairness and Optimal Transport.**   Recent work has drawn many fruitful connection between fairness and optimal transport, mostly by leveraging OT as a tool for obtaining or characterizing fair algorithms [36, 37, 41, 15, 69, 23] — one of the key insights of this line of research being that the problem of finding a fair predictor can be framed as a Wasserstein barycenter problem [37, 16]. Closest to our work is the recent article by Nguyen et al. [56], who consider the constrained problem of finding a Wasserstein barycenter between multiple distributions while controlling the pairwise differences in distances to the barycenter between distributions. Crucially, their setup does not involve any sensitive attribute. We address a different problem, namely obtaining transport plans under mass constraints between groups defined through sensitive attributes. To the best of our knowledge, this problem is still unstudied.

**Constrained Optimal Transport.**   Our work borrows from recent developments in constrained optimal transport, which seeks to enforce structural properties on the transportation plan [17, 6, 58, 50]; for instance, Korman and McCann [46] analyze a variant of optimal transport in which the amount of mass that can be transported between two units is upper bounded. In particular, we build on Rakotomamonjy et al. [64] to obtain finite sample guarantees for a penalized optimal transport problem.

## B   A Modified Sinkhorn Algorithm for Fair Optimal Transport

The following is an immediate consequence of first order conditions on the Lagrangian of Problem 3.

**Proposition B.1.** There exists $\mathbf{f} \in \mathbb{R}^n$, $\mathbf{g} \in \mathbb{R}^m$ and $\mathbf{h} = [h_{sw}]_{sw} \in \mathbb{R}^{K_s \times K_w}$ such that the solution to Problem 3 has the form

$$\mathbf{\Pi} = \mathrm{diag}\big(e^{\mathbf{f}/\varepsilon}\big)\big(\mathbf{K} \odot \mathbf{H}\big)\mathrm{diag}\big(e^{\mathbf{g}/\varepsilon}\big)$$

where

$$\mathbf{K} := \big[e^{-\mathbf{C}_{ij}/\varepsilon}\big]_{ij} \text{ and } \mathbf{H} := \sum_{sw} e^{h_{sw}/\varepsilon}\mathbf{B}_{sw}.$$

**Remark 2.** A similar formulation holds for the cost sensitive problem.

*Proof.* Introducing dual variables $\mathbf{f} \in \mathbb{R}^n, \mathbf{g} \in \mathbb{R}^m$ and $\mathbf{H} = (h_{sw})_{sw} \in \mathbb{R}^{K_s \times K_w}$, the Lagrangian writes

$$
\begin{aligned}
\mathcal{E}\big(\mathbf{\Pi}, \mathbf{f}, \mathbf{g}, \mathbf{h}\big) =& \mathrm{Tr}[\mathbf{\Pi}^\top \mathbf{C}] + \varepsilon \mathbf{KL}(\mathbf{\Pi}) - \mathbf{f}^\top(\mathbf{\Pi}\mathbb{1}_m - \mathbf{a}) - \mathbf{g}^\top(\mathbf{\Pi}^\top\mathbb{1}_n - \mathbf{b}) \\
& - \sum_{s,w} h_{sw}\Big[\mathrm{Tr}\big[\mathbf{\Pi}^\top\mathbf{B}_{sw}\big] - p_s q_w\Big].
\end{aligned}
$$

First order conditions yield for every $i, j$

$$\frac{\partial\mathcal{E}\big(\mathbf{\Pi}, \mathbf{f}, \mathbf{g}, \mathbf{h}\big)}{\partial\mathbf{\Pi}_{ij}} = \mathbf{C}_{ij} + \varepsilon\log\big(\mathbf{\Pi}_{ij}\big) + \varepsilon - \mathbf{f}_i - \mathbf{g}_j - \sum_{sw} h_{sw}\big[\mathbf{B}_{sw}\big]_{ij} = 0$$

which we may rewrite as

$$
\begin{aligned}
\mathbf{\Pi}_{ij} =& \exp\big(\mathbf{f}_i/\varepsilon\big)\exp\Big(-\varepsilon^{-1}\mathbf{C}_{ij} + 1\varepsilon^{-1}\sum_{sw} h_{sw}\big[\mathbf{B}_{sw}\big]_{ij} - 1\Big)\exp\big(\mathbf{g}_j/\varepsilon\big) \\
=& \exp\big(\mathbf{f}_i/\varepsilon\big)\exp(-\mathbf{C}_{ij}/\varepsilon - 1)\prod_{s'=1}^{K_s}\prod_{w'=1}^{K_w}\exp\big(h_{s'w'}[\mathbf{B}_{s'w'}]_{ij}/\varepsilon\big)\exp\big(\mathbf{g}_j/\varepsilon\big) \ .
\end{aligned}
$$

Now, remark that for any $(i, j) \in \{1, \ldots, n\}^2$, there is only one $(s, w) \in \{1, \ldots, K_s\} \times \{1, \ldots, K_w\}$ such that $[\mathbf{B}_{sw}]_{i,j} \neq 0$. Therefore, in the product $\prod_{s'=1}^{K_s}\prod_{w'=1}^{K_w}\exp\big(h_{s'w'}[\mathbf{B}_{s'w'}]_{ij}\big)$, there is only one term distinct from 1. As a consequence

$$
\begin{aligned}
\prod_{s'=1}^{K_s}\prod_{w'=1}^{K_w}\exp\big(h_{s'w'}[\mathbf{B}_{s'w'}]_{ij}/\varepsilon\big) &= \prod_{s'=1}^{K_s}\prod_{w'=1}^{K_w}\exp\big(h_{s'w'}\mathbb{1}_{\mathbf{s}_i=s'}\mathbb{1}_{\mathbf{w}_j=w'}/\varepsilon\big) \\
&= \sum_{s'=1}^{K_s}\sum_{w'=1}^{K_w}\exp\big(h_{s'w'}/\varepsilon\big)\mathbb{1}_{\mathbf{s}_i=s'}\mathbb{1}_{\mathbf{w}_j=w'} \\
&= \sum_{s'=1}^{K_s}\sum_{w'=1}^{K_w}\exp\big(h_{s'w'}/\varepsilon\big)\big[\mathbf{B}_{s'w'}\big]_{ij} \ .
\end{aligned}
$$

This leads to the matrix form

$$\mathbf{\Pi} = \mathrm{diag}\big(e^{\mathbf{f}/\varepsilon}\big)\big(\mathbf{K} \odot \mathbf{H}\big)\mathrm{diag}\big(e^{\mathbf{g}/\varepsilon}\big)$$

where

$$\mathbf{K} := e^{-\mathbf{C}/\varepsilon - 1}$$

$$\mathbf{H} := \sum_{s=1}^{K_s}\sum_{w=1}^{K_w} e^{h_{sw}/\varepsilon}\mathbf{B}_{sw}.$$

$\square$

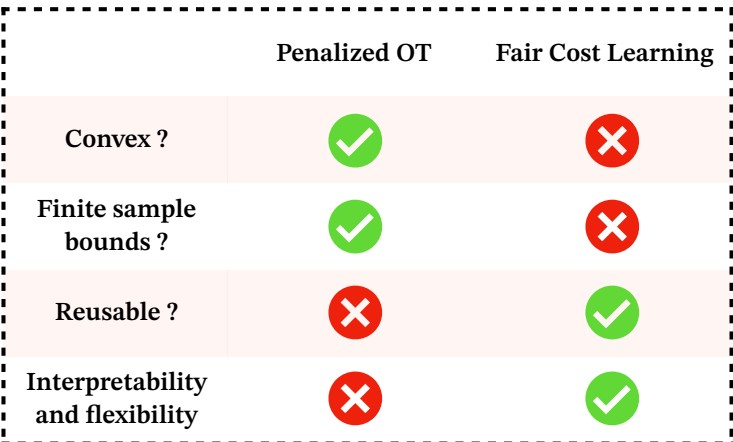

| | Penalized OT | Fair Cost Learning |
|---|:---:|:---:|
| Convex ? | ✅ | ❌ |
| Finite sample bounds ? | ✅ | ❌ |
| Reusable ? | ❌ | ✅ |
| Interpretability and flexibility | ❌ | ✅ |

Figure 2: Comparison of both approaches.

A fair optimal transport plan can hence be obtained using a modified Sinkhorn algorithm which we call `FairSinkhorn`. Define the map

$$\Phi(\mathbf{u}, \mathbf{v}) := \left( \mathrm{Tr}\left[ \left( \left[ \mathbf{u}_i \mathbf{v}_j \right]_{ij} \odot \mathbf{B}_{sw} \right) \left( \mathbf{K} \odot \mathbf{B}_{sw} \right)^\top \right] \right)_{sw}.$$

The algorithm `FairSinkhorn` combines projections on the set of marginal constraints with a projection on the set of fairness constraints, highlighted in blue.

---

**Algorithm 1** `FairSinkhorn` Algorithm

---

1: **Inputs:** Cost $\mathbf{C} \in \mathbb{R}^{n \times m}$, $\varepsilon > 0$, marginals $(\mathbf{a}, \mathbf{b}) \in \mathbb{R}_+^n \times \mathbb{R}_+^m$, fairness target $\mathbf{F} = (\mathbf{F}_{sw})_{sw} \in [0,1]^{K_s \times K_w}$, max iter. $T$.
2: $\mathbf{K} \leftarrow e^{-\mathbf{C}/\varepsilon - 1}$
3: **for** $t = 1, \ldots, T$ **do**
4:     $\mathbf{u}^{(t+1)} \leftarrow \mathbf{a} \oslash \left( (\mathbf{K} \odot \mathbf{T}^{(t)}) \mathbf{v}^{(t)} \right)$
5:     $\mathbf{v}^{(t+1)} \leftarrow \mathbf{b} \oslash \left( (\mathbf{K} \odot \mathbf{T}^{(t)})^\top \mathbf{u}^{(t+1)} \right)$
6:     $\mathbf{L}^{(t+1)} \leftarrow \mathbf{F} \oslash \Phi(\mathbf{u}^{(t+1)}, \mathbf{v}^{(t+1)})$
7:     $\mathbf{T}^{(t+1)} \leftarrow \sum_{sw} \ell_{sw}^{(t+1)} \mathbf{B}_{sw}$ w. $\mathbf{L}^{(t+1)} = \left( \ell_{sw}^{(t+1)} \right)_{sw}$
8: **end for**
9: **return**
$$\mathbf{\Pi} = \mathrm{diag}\left( \mathbf{u}^{(T+1)} \right) \left( \mathbf{K} \odot \mathbf{T}^{(T+1)} \right) \mathrm{diag}\left( \mathbf{v}^{(T+1)} \right)$$

---

**Remark 3.** Since `FairSinkhorn` alternates between projections on sets defined by linear constraints, it is guaranteed to converge [5].

## C Which approach should one choose ?

**Convexity.** While the penalized OT problem is convex, the fair cost learning approach requires bi-level optimization — hence, the solution is not guaranteed to be unique and algorithms might not converge easily.

**Theoretical Guarantees.** Our first approach comes with strong theoretical guarantees, which ensure a convergence towards the value of the population problem at rate $n^{-1/2}$. The rate of convergence of our second approach remains an open problem.

**Reusability.** When using the first approach, one need to solve a new OT problem for every new batch of samples. Interestingly, once a cost function that enforces fair matchings is learned, it can be reuse on any set of points. This also ensures lower variance of fairness metrics (see experiments).

395 **Interpretability and flexibility.** Our second approach learns a parametrized cost function which
396 can be interpreted and used for subsequent downstream tasks. It also offers greater flexibility, since
397 the baseline cost function and the discrepancy $\mathscr{D}$ can be set by the user.

# D Proofs

## D.1 Proof of Proposition 2.4

**Lemma D.1.** Let $\mu \in \mathcal{P}(\mathcal{X} \times \mathcal{S})$ and $\nu \in \mathcal{P}(\mathcal{Y} \times \mathcal{W})$. Let $p \in \mathcal{P}(\mathcal{S})$ and $q \in \mathcal{P}(\mathcal{W})$ be obtained
from $\mu$ and $\nu$ by marginalizing, respectively, $x$ and $y$, that is,

$$p(S = s) = \mu(\mathcal{X} \times \{s\}) \tag{13}$$
$$q(W = w) = \nu(\mathcal{Y} \times \{w\}). \tag{14}$$

Finally let $F \in \Pi(p, q)$.

There exists $\pi \in \Pi(\mu, \nu)$ such that $\pi$ is $F-$fair, that is

$$\pi\big(\mathcal{X} \times \{s\} \times \mathcal{Y} \times \{w\}\big) = F(S = s, W = w).$$

*Proof.* Given measurable sets $A \subseteq \mathcal{X}$ and $B \subseteq \mathcal{Y}$ let $\pi$ be defined as

$$\pi\big(A \times \{s\} \times B \times \{w\}\big) := \frac{F(S = s, W = w)\mu(A \times \{s\})\nu(B \times \{w\})}{p(S = s)q(W = w)}$$

and let's check that **i)** $\pi \in \Pi(\mu, \nu)$ and **ii)** $\pi$ is $F-$fair:

    i) We want to check that $\pi(A \times \{s\} \times \mathcal{Y} \times \mathcal{W}) = \mu(A \times \{s\})$ and similarly for the other
    marginal. To this end note that

$$\pi(A \times \{s\} \times \mathcal{Y} \times \mathcal{W}) = \sum_{w \in \mathcal{W}} \frac{F(S = s, W = w)\mu(A \times \{s\})\nu(\mathcal{Y} \times \{w\})}{p(S = s)q(W = w)}$$
$$= \mu(A \times \{s\}) \sum_{w \in \mathcal{W}} \frac{F(S = s, W = w)}{p(S = s)},$$

    where the second equality follows from (14). To finish note that from $F \in \Pi(p, q)$ follows
    that $\sum_{w \in \mathcal{W}} F(S = s, W = w) = p(S = s)$. A similar argument shows that $\pi(\mathcal{X} \times \mathcal{S} \times$
    $B \times \{w\}) = \nu(B \times \{w\})$.

    ii) To see that $\pi$ is $F-$fair note that it is immediate from (13) and (14) that

$$\pi\big(\mathcal{X} \times \{s\} \times \mathcal{Y} \times \{w\}\big) = \frac{F(S = s, W = w)\mu(\mathcal{X} \times \{s\})\nu(\mathcal{Y} \times \{w\})}{p(S = s)q(W = w)}$$
$$= F(S = s, W = w).$$

$\square$

**Proposition D.1.** *Assume that $\mu$ and $\eta$ have bounded support. Let $F$ be a coupling of the marginals
of $\mu$ and $\nu$ denoted by $p$ and $q$ defined via (13)-(14). Then there exists a unique $F$-fair transport plan.*

*Proof.* Assume $\mathcal{X}$ and $\mathcal{Y}$ to be compact. Similar to the proof of Theorem 1.4. in [66] we can prove
that

$$\Lambda := \big\{\pi \in \mathcal{P}(\mathcal{X} \times \{0, 1\} \times \mathcal{Y} \times \{0, 1\}) : \pi_{SW} = F\big\}$$

is compact with respect to the weak topology: Let $\pi_n$ be a sequence in $\Lambda$. They are probability
measures, so that their mass is 1, and hence they are bounded in the dual of $C(\mathcal{X} \times \{0, 1\} \times \mathcal{Y} \times \{0, 1\})$.
Hence usual weak-$\star$ compactness in dual spaces guarantees the existence of a subsequence $\pi_n \rightharpoonup \pi$
converging to a probability $\pi$. We just need to check $\pi \in \Lambda$. This may be done by fixing $\phi \in$
$C(\{0, 1\} \times \{0, 1\})$ and from $\pi_n \in \Lambda$ it follows that

$$\int \phi \, d\pi_n = \int \phi \, d\big[(\pi_n)_{SW}\big] = \int \phi \, dF = \sum_{s,w} \phi(s, w)F(S = s, W = w).$$

Now pass to the limit to obtain

$$\int \phi \, d\pi_{SW} = \int \phi \, d\pi = \sum_{s,w} \phi(s,w) F(S = s, W = W)$$

422  which shows that $\pi \in \Lambda$.

423  To finish, just note that $\Pi_{\text{fair}}^{\mathbf{F}} = \Pi(\mu, \nu) \cap \Lambda$ is the intersection of two compact sets (with respect to
424  weak topology) and Lemma D.1 establishes that it is non-empty. Continuity of the map defining
425  the transport problem is enough to conclude the existence of a minimizer. Uniqueness is then a
426  consequence of the strict convexity of KL. □

## D.2   Proof of Theorem 3.1

428  **Notations.**   Let $\mathbf{F} \in \mathbb{R}^{K_s \times K_w}$ a given a fairness target. In what follows, we focus on $K_s = K_w = 2$,
429  and adopt the following notational conventions.

$$\begin{aligned}
\xi_{11}(u,t) &:= ut - \mathbf{F}_{11} \quad \xi_{10}(u,t) := u(1-t) - \mathbf{F}_{10}, \\
\xi_{01}(u,t) &:= (1-u)t - \mathbf{F}_{01}, \quad \xi_{00}(u,t) := (1-u)(1-t) - \mathbf{F}_{00}.
\end{aligned}$$

430  Given a measure $\pi \in \mathcal{M}(\mathcal{X} \times \{0,1\} \times \mathcal{Y} \times \{0,1\})$, define

$$\mathcal{L}_{\mathbf{F}}(\pi) := \sum_{s,w \in \{0,1\}^2} \langle \xi_{sw}, \pi \rangle^2 = \sum_{s,w \in \{0,1\}^2} \left( \int \xi_{sw}(z,w) \, d\pi(x,z,y,w) \right)^2.$$

431  For any distribution $\alpha$, $\beta$ over a set $\mathcal{X}$ such that $\alpha$ is absolutely continuous with respect to $\beta$ (i.e.,
432  $\alpha \ll \beta$), we denote by $\frac{d\alpha}{d\beta}$ the Radon-Nikodym derivative of $\alpha$ with respect to $\beta$, and

$$\mathbf{KL}(\alpha || \beta) := \int_{\mathcal{X}} \log \left( \frac{d\alpha}{d\beta}(x) \right) d\alpha(x).$$

433  Given any two measures $\alpha \in \mathcal{M}(\mathcal{X} \times \{0,1\})$ and $\beta \in \mathcal{M}(\mathcal{Y} \times \{0,1\})$ define

$$m^{\star}(\alpha, \beta) := \min_{\pi \in \Pi(\alpha, \beta)} \langle c, \pi \rangle + \varepsilon \mathbf{KL}(\pi || \alpha \otimes \beta) + \lambda \mathcal{L}_{\mathbf{F}}(\pi). \tag{15}$$

434  The goal is to prove the following result

435  **Theorem D.2.**   Let $\mathcal{X}$ and $\mathcal{Y}$ be compact subsets of $\mathbb{R}^d$ and let $\mu$ and $\eta$ be probability measures on
436  $\mathcal{X} \times \{0,1\}$ and $\mathcal{Y} \times \{0,1\}$, respectively. Let $(x_i, s_i)_{i=1}^n$ be $n$ independent and identically distributed
437  (i.i.d.) samples from $\mu$ and let $(y_j, w_j)_{j=1}^n$ be $n$ i.i.d. samples from $\eta$; assume further that the samples
438  from $\mu$ are independent from those from $\eta$. Finally, suppose that the cost $c$ is $L$-Lipschitz and $\mathcal{C}^{\infty}$.
439  Then

$$\mathbb{E}_{\mu \otimes \eta} |m^{\star}(\mu_n, \eta_n) - m^{\star}(\mu, \eta)| \leq \mathcal{O}(\log(n)/\sqrt{n}^{-1}), \tag{16}$$

440  *Proof.*   Let $m^{\star}_{\infty} := m^{\star}(\mu, \eta)$ and $m^{\star}_n := m^{\star}(\mu_n, \eta_n)$, and $\pi^{\star}_{\infty}$ be the minimizer attaining the $m^{\star}_{\infty}$
441  that is a measure on $\mathcal{X} \times \{0,1\} \times \mathcal{Y} \times \{0,1\}$ with marginals $\mu$ and $\eta$ such that

$$m^{\star}_{\infty} = \langle c, \pi^{\star}_{\infty} \rangle + \varepsilon \mathbf{KL}(\pi^{\star}_{\infty} || \mu \otimes \eta) + \lambda \mathcal{L}_{\mathbf{F}}(\pi^{\star}_{\infty}).$$

442  Let $p^{\star}_{\infty}$ denote the Radon–Nikodym density of $\pi^{\star}_{\infty}$ with respect to $\mu \otimes \eta$ (note that the Kull-
443  back–Leibler term forces $\pi^{\star}_{\infty}$ to be absolutely continuous with respect to $\mu \otimes \eta$).

444  The idea is to "sandwich" the random variable $m^{\star}_n - m^{\star}_{\infty}$ between two random variables with
445  expectation upper bounded by $\mathcal{O}(\log(n)/\sqrt{n})$, that is, to find random variables $Y_n, Z_n$ such that

$$Y_n - m^{\star}_{\infty} \leq m^{\star}_n - m^{\star}_{\infty} \leq Z_n - m^{\star}_{\infty} \tag{17}$$

446  and

$$\mathbb{E}[|Y_n - m^{\star}_{\infty}|] \leq \mathcal{O}(\log(n)/\sqrt{n}^{-1}) \quad \text{and} \quad \mathbb{E}[|Z_n - m^{\star}_{\infty}|] \leq \mathcal{O}(\log(n)/\sqrt{n}^{-1}).$$

447  To see that this is enough to establish (16), observe that (17) implies that

$$|m^{\star}_n - m^{\star}_{\infty}| \leq \max(|Y_n - m^{\star}_{\infty}|, |Z_n - m^{\star}_{\infty}|) \leq |Y_n - m^{\star}_{\infty}| + |Z_n - m^{\star}_{\infty}|.$$

**Lower bound.** To define $Y_n$ start by observing that by linearizing the fairness penalization, we can incorporate this linearization into the transport cost, so that the problem is now a proper entropic optimal transport one. This allows us to leverage existing results on the sample complexity of optimal transport. Moreover, the convexity of $\mathcal{L}_\mathbf{F}$ implies that this linearization does yield a lower bound. To this end note that for any measure $\pi \in \mathcal{M}(\mathcal{X} \times \{0,1\} \times \mathcal{Y} \times \{0,1\})$ and any $s, w \in \{0,1\}^2$, the inequality $0 \leq (\langle \xi_{sw}, \pi - \pi_\infty^\star \rangle)^2 = (\langle \xi_{sw}, \pi \rangle)^2 + (\langle \xi_{sw}, \pi_\infty^\star \rangle)^2 - 2\langle \xi_{sw}, \pi \rangle \langle \xi_{sw}, \pi_\infty^\star \rangle$, implies that

$$\mathcal{L}_\mathbf{F}(\pi) \geq -\mathcal{L}_\mathbf{F}(\pi_\infty^\star) + 2 \sum_{(s,w)\in\{0,1\}^2} \langle \langle \xi_{sw}, \pi_\infty^\star \rangle \xi_{sw}, \pi \rangle \tag{18}$$

$$= \mathcal{L}_\mathbf{F}(\pi_\infty^\star) + \Big\langle 2 \sum_{s,w\in\{0,1\}^2} \langle \xi_{sw}, \pi_\infty^\star \rangle \xi_{sw}, \pi - \pi_\infty^\star \Big\rangle. \tag{19}$$

This leads to a lower bound on $m_n^\star$ given by

$$m_n^\star \geq -\lambda\mathcal{L}_\mathbf{F}(\pi_\infty^\star) + \underbrace{\min_{\substack{\pi_1=\mu_n\\\pi_2=\eta_n}} \Big\langle \overbrace{c + 2\lambda \sum_{s,w\in\{0,1\}^2} \langle \xi_{sw}, \pi_\infty^\star \rangle \xi_{sw}}^{:=\widehat{c}}, \pi \Big\rangle + \varepsilon\mathbf{KL}(\pi || \mu_n \otimes \eta_n)}_{:=\widehat{m}_n^\star} := Y_n \tag{20}$$

where $\widehat{m}_n^\star$ is an entropic optimal transport problem with cost $\widehat{c}$. From the sample complexity of optimal transport (see Theorem 18 in [33]) it follows that

$$\mathbb{E}_{\mu\otimes\eta}|\widehat{m}_n^\star - \widehat{m}_\infty^\star| \leq \mathcal{O}(\sqrt{n}^{-1}), \tag{21}$$

with $\widehat{m}_\infty^\star$ being the population version of $\widehat{m}_n^\star$, *i.e.*,

$$\widehat{m}_\infty^\star := \min_{\substack{\pi_1=\mu\\\pi_2=\eta}} \Big\langle c + 2\lambda \sum_{s,w\in\{0,1\}^2} \langle \xi_{sw}, \pi_\infty^\star \rangle \xi_{sw}, \pi \Big\rangle + \varepsilon\mathbf{KL}(\pi || \mu \otimes \eta).$$

To finish the lower bound, note that $\mathbb{E}[|Y_n - m_\infty^\star|] \leq \mathcal{O}(\sqrt{n}^{-1})$ follows from (21) provided we show that $m_\infty^\star = \widehat{m}_\infty^\star - \lambda\mathcal{L}_\mathbf{F}(\pi_\infty^\star)$; this is the content of Lemma D.4 that can be found in Appendix D.2.2.

**Upper bound.** Define the random positive measure

$$\widehat{\pi}_n^\star = \frac{1}{n^2} \sum_{ij} p_\infty^\star((x_i, s_i), (y_j, w_j))\delta_{((x_i,s_i),(y_j,w_j))} \tag{22}$$

and define $\widehat{Z}_n$ by

$$\widehat{Z}_n := \langle c, \widehat{\pi}_n^\star \rangle + \varepsilon\mathbf{KL}(\widehat{\pi}_n^\star || \mu_n \otimes \eta_n) + \lambda\mathcal{L}_\mathbf{F}(\widehat{\pi}_n^\star).$$

Observe that $\widehat{\pi}_n^\star$ does not necessarily satisfies the marginal constraints that define $\Pi(\mu, \eta)$, and, consequently, $\widehat{Z}_n$ is not necessarily an upper bound of $m_n^\star$. A natural idea is to project $\widehat{\pi}_n^\star$ onto the constraint set. In what follows we interchangeably use the notation $\widehat{\pi}_n^\star$ to denote (22) and the matrix with $ij$ entry given by $p_\infty^\star((x_i, s_i), (y_j, w_j))/n^2$. An interesting projection to consider is the one proposed in [1, Algorithm 2] named *Round*, reviewed in Appendix D.2.2. Let

$$\bar{\pi}_n^\star = \mathrm{Round}(\widehat{\pi}_n^\star),$$

where, similar to $\widehat{\pi}_n^\star$, the notation $\bar{\pi}_n^\star$ denotes both the matrix and the measure $\sum_{ij}(\bar{\pi}_n^\star)_{ij}\delta_{(x_i,s_i),(y_j,w_j)}$. Define

$$Z_n := \int c \, d\bar{\pi}_n^\star + \varepsilon\mathbf{KL}(\bar{\pi}_n^\star || \mu_n \otimes \eta_n) + \lambda\mathcal{L}_\mathbf{F}(\bar{\pi}_n^\star). \tag{23}$$

By construction, $\bar{\pi}_n^\star$ satisfies the marginal constraints, hence it holds that $Z_n \geq m_n^\star$. The remaining goal is thus to prove $\mathbb{E}[|Z_n - m^\star|] \leq \mathcal{O}(\log(n)/\sqrt{n})$. To do so the idea is to compare $Z_n$ with $\widehat{Z}_n$ and $\widehat{Z}_n$ with $m_\infty^\star$. In fact, from

$$|Z_n - m_\infty^\star| \leq |Z_n - \widehat{Z}_n| + |\widehat{Z}_n - m_\infty^\star|,$$

it is sufficient to prove *(i)* $\mathbb{E}|Z_n - \widehat{Z}_n| \in \mathcal{O}(\log(n)/\sqrt{n}^{-1})$ and *(ii)* $\mathbb{E}|\widehat{Z}_n - m_\infty^\star| \in \mathcal{O}(\log(n)/\sqrt{n}^{-1})$.

*(i).* $\mathbb{E}|\widehat{Z}_n - m_\infty^\star|$. Let $V_{i,j} := ((x_i, s_i), (y_j, w_j))$ and since the samples are i.i.d.,

$$\mathbb{E}\langle c, \hat{\pi}_n^\star \rangle = \frac{1}{n^2} \sum_{ij} \mathbb{E}_{\mu \otimes \eta} [p_\infty^\star(V_{i,j}) c(V_{i,j})] = \langle cp_\infty^\star, \mu \otimes \eta \rangle = \langle c, \pi_\infty^\star \rangle.$$

Moreover, let $d\hat{\pi}_n^\star/d(\mu_n \otimes \eta_n)$ be the Radon-Nikodym derivative of $\hat{\pi}_n^\star$ with respect to $\mu_n \otimes \eta_n$ and, by definition,

$$\mathbb{E}\left[\mathbf{KL}(\hat{\pi}_n^\star||\mu_n \otimes \eta_n)\right] = \mathbb{E}\langle \log \frac{d\hat{\pi}_n^\star}{d(\mu_n \otimes \eta_n)}, \hat{\pi}_n^\star \rangle = \frac{1}{n^2} \sum_{ij} \mathbb{E}_{\mu \otimes \eta} \log p_\infty^\star(V_{i,j}) \, p_\infty^\star(V_{i,j})$$

$$= \langle \log(p_\infty^\star) p_\infty^\star, \mu \otimes \eta \rangle = \langle \log \frac{d\pi_\infty^\star}{d(\mu \otimes \eta)}, \pi_\infty^\star \rangle = \mathbf{KL}(\pi_\infty^\star||\mu \otimes \eta).$$

Since $\mathcal{X}$ and $\mathcal{Y}$ are compact and $c$ is continuous it follows that $c(V_{i,j})$ is uniformly bounded. Furthermore, by Lemma D.6, there exists $r > 0$ such that, $\mu \otimes \eta$-a.s., $\frac{1}{r} \leq p_\infty^\star(V_{i,j}) \leq r$. Consequently, it follows by Lemma D.5,

$$\max\{\mathbb{E}|\langle c, \hat{\pi}_n^\star - \pi_\infty^\star \rangle|, \mathbb{E}|\mathbf{KL}(\hat{\pi}_n^\star||\mu_n \otimes \eta_n) - \mathbf{KL}(\pi_\infty^\star||\mu \otimes \eta)|\} \in \mathcal{O}(\sqrt{n}^{-1}). \quad (24)$$

Finally, for the last term, note that $x^2 - y^2 = (x - y)^2 + 2y(x - y)$, hence for $s, w \in \{0, 1\}$

$$\mathbb{E}|\langle \xi_{sw}, \hat{\pi}_n^\star \rangle^2 - \langle \xi_{sw}, \pi_\infty^\star \rangle^2| \leq \mathbb{E}\langle \xi_{sw}, \hat{\pi}_n^\star - \pi_\infty^\star \rangle^2 + 2\mathbb{E}\left[|\langle \xi_{sw}, \pi_\infty^\star \rangle \langle \xi_{sw}, \hat{\pi}_n^\star - \pi_\infty^\star \rangle|\right] \in \mathcal{O}(\sqrt{n}^{-1}), \quad (25)$$

where we conclude similarly as above using Lemma D.5.

Inequalities (24) and (25) are enough to establish $\mathbb{E}[|\widehat{Z}_n - m_\infty^\star|] \in \mathcal{O}(\sqrt{n}^{-1})$ and conclude the proof of *(i)*.

*(ii).* $\mathbb{E}|Z_n - \widehat{Z}_n|$. To obtain a bound on $\mathbb{E}|Z_n - \widehat{Z}_n|$ we first translate it into a bound on $\mathbb{E}\|\hat{\pi}_n^\star - \bar{\pi}_n^\star\|_1$. In fact, Lemma D.7 shows that

$$\mathbb{E}|Z_n - \widehat{Z}_n| \leq \mathcal{O}(\log(n))\mathbb{E}\|\hat{\pi}_n^\star - \bar{\pi}_n^\star\|_1$$

Moreover [1, Lemma 7] implies that

$$\|\hat{\pi}_n^\star - \bar{\pi}_n^\star\|_1 \leq 2\|\hat{\pi}_n^\star \mathbf{1} - 1/n\mathbf{1}\|_1 + 2\|(\hat{\pi}_n^\star)^T \mathbf{1} - 1/n\mathbf{1}\|_1, \quad (26)$$

and, consequently, to obtain the bound on $\mathbb{E}|Z_n - \widehat{Z}_n|$ it is enough to show that, in expectation, the right-hand-side of (26) is $\mathcal{O}(\sqrt{n}^{-1})$. This last bound essentially reduces to the proof of Lemma 14 of [65] together with the Cauchy Schwartz inequality – this is made precise in Lemma D.8. $\qquad\square$

### D.2.1 Review of Round

This section reviews the Round map of [1] defined as: Given $M \in \mathbb{R}_{\geq 0}^{n^2}$, the matrix $G := \text{Round}(M)$ is obtained via

1. Let $M' := \text{diag}(x)M$, with $x_i := \min\left(1, \frac{1/n}{r_i(M)}\right)$ and $r_i(M) := \left(M\mathbf{1}\right)_i$

2. Let $M'' := M'\text{diag}(y)$, with $y_j := \min\left(1, \frac{1/n}{c_j(M')}\right)$ and $c_j(M') := \left((M')^T\mathbf{1}\right)_j$

3. Output $G := M'' + \frac{1}{\|\text{err}_r\|_1}\text{err}_r\text{err}_c^T$, where

$$\text{err}_r := 1/n\mathbf{1} - r(M'') \quad \text{and} \quad \text{err}_c := 1/n\mathbf{1} - c(M'').$$

**Lemma D.3.** Suppose that $\exp(-K)/n^2 \leq M_{ij} \leq \exp(K)/n^2$ and let $G := \text{Round}(M)$. Then $G_{ij} \geq \exp(-4K)/n^2$.

*Proof.* First observe that

**Claim 1.** $\exp(-2K)/n^2 \le M'_{ij} \le M_{ij} \le \exp(K)/n^2$.

To see this let $M_i$ denote the $i$th line of $M$ and similarly for $M'$. We have that

$$M'_i = x_i M_i \le M_i, \quad \text{since} \quad x_i \le 1.$$

which shows that $M'_{ij} \le M_{ij} \le \exp(K)/n^2$.

To obtain the lower bound suppose that $x_i < 1$ (otherwise there is nothing to prove). In this case we have $M'_i = \frac{1/n}{r_i(M)} M_i$ and, from the upper bounds on $M$,

$$r_i(M) = \sum_j M_{ij} \le n(\exp(K)/n^2).$$

As a consequence,

$$M'_{ij} \ge \frac{1/n}{n(\exp(K)/n^2)} M_{ij} = \exp(-K) M_{ij} \ge \exp(-2K)/n^2,$$

thus establishing claim 1.

**Claim 2.** $\exp(-4K)/n^2 \le M''_{ij} \le M'_{ij} \le \exp(K)/n^2$.

To prove Claim 2 we argue as in Claim 1.

**Claim 3.** The vectors $\text{err}_r$ and $\text{err}_c$ are both non-negative.

To see this just observe that after step 1, row $i$ of $M'$ satisfies $r_i(M') \le 1/n$ and, since $\text{diag}(y) \le \text{diag}(\mathbf{1})$, step 2 can only decrease $r_i$. A similar reasoning shows that $c_j(M'') \le 1/n$. $\square$

### D.2.2 Technical Lemmas

**Lemma D.4.** Let $\pi_\infty^\star$ be a minimizer of

$$\min_{\substack{\pi_1 = \mu \\ \pi_2 = \eta}} F(\pi) := \langle c, \pi \rangle + \varepsilon \mathbf{KL}\big(\pi \| \mu \otimes \eta\big) + \lambda \mathcal{L}_{\mathbf{F}}(\pi) := m_\infty^\star. \tag{27}$$

Then $\pi_\infty^\star$ also minimizes

$$\min_{\substack{\pi_1 = \mu \\ \pi_2 = \eta}} F_L(\pi) := \langle c, \pi \rangle + \varepsilon \mathbf{KL}(\pi \| \mu \otimes \eta) + \lambda \mathcal{L}_{\mathbf{F}}(\pi_\infty^\star) + \Big\langle 2\lambda \sum_{s,w \in \{0,1\}^2} \langle \xi_{sw}, \pi_\infty^\star \rangle \xi_{sw}, \pi - \pi_\infty^\star \Big\rangle \tag{28}$$

$$= \min_{\substack{\pi_1 = \mu \\ \pi_2 = \eta}} \Big\langle c + 2\lambda \sum_{s,w \in \{0,1\}^2} \langle \xi_{sw}, \pi_\infty^\star \rangle \xi_{sw}, \pi \Big\rangle + \varepsilon \mathbf{KL}(\pi \| \mu \otimes \eta) - \lambda \mathcal{L}_{\mathbf{F}}(\pi_\infty^\star)$$

$$= \widehat{m}_\infty^\star - \lambda \mathcal{L}_{\mathbf{F}}(\pi_\infty^\star).$$

and, consequently, $\widehat{m}_\infty^\star - \lambda \mathcal{L}_{\mathbf{F}}(\pi_\infty^\star) = m_\infty^\star$.

*Proof.* The proof is inspired by the one from [64, Proposition 2.1]. Assume $\pi_\infty^\star$ minimizes Eq. (27). The presence of entropic regularization implies that $\pi_\infty^\star \ll \mu \otimes \eta$. Let

$$g_L = 2 \sum_{s,w \in \{0,1\}^2} \langle \xi_{sw}, \pi_\infty^\star \rangle \xi_{sw}.$$

We already know that for any $\pi \in \Pi(\mu, \eta)$

$$\mathcal{L}_{\mathbf{F}}(\pi) \ge \mathcal{L}_{\mathbf{F}}(\pi_\infty^\star) + \langle g_L, \pi - \pi_\infty^\star \rangle.$$

Define

$$g_{\mathrm{KL}} = \log\left( \frac{d\pi_\infty^\star}{d(\mu \otimes \eta)} \right),$$

we prove, for any $\pi \in \Pi(\mu, \eta)$,

$$\mathbf{KL}(\pi \| \mu \otimes \eta) \ge \mathbf{KL}(\pi_\infty^\star \| \mu \otimes \eta) + \langle g_{\mathrm{KL}}, \pi - \pi_\infty^\star \rangle,$$

with convex analysis[1], and thus

$$F(\pi) \geq F(\pi_\infty^\star) + \langle c + \lambda g_L + \varepsilon g_{\mathrm{KL}}, \pi - \pi_\infty^\star \rangle.$$

Consequently, for any $\pi \in \Pi(\mu, \eta)$ such that $\pi \ll \mu \otimes \eta^2$,

$$\langle c + \lambda g_L + \varepsilon g_{\mathrm{KL}}, \pi - \pi_\infty^\star \rangle \geq 0,$$

otherwise, for $r \in [0, 1]$, $\pi_r = (1-r)\pi_\infty^\star + r\pi \in \Pi(\mu, \eta) \cap \{\nu; \nu \ll \mu \otimes \eta\}$, we could find $r^\star \in (0, 1]$, such that $F(\pi_{r^\star}) < F(\pi_\infty^\star)$. In fact, by contradiction, assume $\langle c + \lambda g_L + \varepsilon g_{\mathrm{KL}}, \pi - \pi_\infty^\star \rangle < 0$, as $\lim_{r \to 0^+} \frac{F(\pi_r) - F(\pi_\infty^\star)}{r} = \langle c + \lambda g_L + \varepsilon g_{\mathrm{KL}}, \pi - \pi_\infty^\star \rangle$, we have, by continuity, the existence of a sufficiently small $r^\star \in (0, 1]$ such that $F(\pi_{r^\star}) < F(\pi_\infty^\star)$. Moreover, as $F_L(\pi_\infty^\star) = F(\pi_\infty^\star)$, for any $\pi \in \Pi(\mu, \eta)$

$$F_L(\pi) \geq F_L(\pi_\infty^\star) + \langle c + \lambda g_L + \varepsilon g_{\mathrm{KL}}, \pi - \pi_\infty^\star \rangle \geq F_L(\pi_\infty^\star).$$

Hence, $\pi_\infty^\star$ is also a minimizer of Eq. (28). $\qquad\square$

**Lemma D.5.** Suppose $a \in L^\infty(\mu \otimes \eta)$ is such that $\langle a, \mu \otimes \eta \rangle = 0$. Then,

$$\mathbb{E}\,|\langle a, \mu_n \otimes \eta_n \rangle| \in \mathcal{O}(\sqrt{n}^{-1}) \quad \text{and} \quad \mathbb{E}\langle a, \mu_n \otimes \eta_n \rangle^2 \in \mathcal{O}(n^{-1}),$$

*Proof.* Let $\mathfrak{S}_n$ denote the set of all permutation on $n$ elements. By counting terms, we observe that

$$\langle a, \mu_n \otimes \eta_n \rangle = \frac{1}{n!} \sum_{\sigma \in \mathfrak{S}_n} \frac{1}{n} \sum_{k=1}^n a((X_k, S_k), (Y_{\sigma(k)}, W_{\sigma(k)})).$$

In fact, there are $(n-1)!$ permutations $\sigma \in \mathfrak{S}_n$ satisfying $\sigma(i) = j$ because there are $(n-1)!$ ways to permute the rest. Hence, the contribution of the specific $a((X_k, S_k), (Y_{\sigma(k)}, W_{\sigma(k)}))$ is weighted by $\frac{1}{n!} \frac{(n-1)!}{n} = \frac{1}{n^2}$. Now, we observe that for any fixed permutation $\sigma$, the joint law $((X_1, S_1), (Y_{\sigma(1)}, W_{\sigma(1)})), \ldots, ((X_n, S_n), (Y_{\sigma(n)}, W_{\sigma(n)}))$ is identical to that of $V_1, \ldots, V_n$ where $V_k \sim \mu \otimes \eta$ are independent and identically distributed. Thus,

$$\mathbb{E}\left[\langle a, \mu_n \otimes \eta_n \rangle\right] \leq \frac{1}{n!} \sum_{\sigma \in \mathfrak{S}_n} \frac{1}{n} \mathbb{E}\left|\sum_{k=1}^n a(V_k)\right| = \frac{1}{n}\mathbb{E}\left|\sum_{k=1}^n a(V_k)\right| \leq \frac{1}{n}\sqrt{\mathbb{E}\left[\left(\sum_{k=1}^n a(V_k)\right)^2\right]}$$

$$= \frac{1}{n}\sqrt{\mathbb{E}\left[\sum_{k=1}^n a(V_k)^2\right]} \leq \frac{\|a\|_{L^\infty(\mu \otimes \eta)}}{\sqrt{n}}.$$

Moreover, by applying Jensen inequality,

$$\mathbb{E}\langle a, \mu_n \otimes \eta_n \rangle^2 \leq \frac{1}{n!} \sum_{\sigma \in \mathfrak{S}_n} \mathbb{E}\left(\frac{1}{n}\sum_{k=1}^n a(V_k)\right)^2 = \mathbb{E}\left(\frac{1}{n}\sum_{k=1}^n a(V_k)\right)^2 \leq \frac{1}{n^2}\mathbb{E}\left(\sum_{k=1}^n a(V_k)\right)^2$$

$$\leq \frac{1}{n^2}\mathbb{E}\sum_{k=1}^n a(V_k)^2 \leq \frac{\|a\|_{L^\infty(\mu \otimes \eta)}^2}{n}. \quad\square$$

**Lemma D.6.** If $\mathcal{X}$ and $\mathcal{Y}$ are compact, then there exists $r, s > 0$[3] such that, $\mu \otimes \eta$-a.s.,

$$\frac{1}{r} \leq p_\infty^\star(V_{i,j}) \leq r, \quad \text{and} \quad \frac{1}{s} \leq \frac{d\bar{\pi}_n^\star}{d(\mu_n \otimes \eta_n)} \leq n^2.$$

---

[1]If $\pi$ is singular with respect to $\mu \otimes \eta$, the inequality is trivial. Assume $\pi \ll \mu \otimes \eta$, we define for $r \in [0, 1]$, $\pi_r = (1 - r)\pi_\infty^\star + r\pi \in \Pi(\mu, \eta)$ and $\phi(r) = \mathbf{KL}(\pi_r \| \mu \otimes \eta)$. By convexity of $\phi$, $\phi(r) \geq \phi(0) + r \lim_{h \to 0^+} \frac{\phi(h) - \phi(0)}{h}$. We compute the previous directional limit using the monotone convergence theorem as in Csiszár [19, Lemma 2.1].

[2]If instead $\pi$ is singular with respect to $\mu \otimes \eta$, then $F(\pi) = F_L(\pi) = \infty$, so such measures are trivially excluded from being minimizers of either functional.

[3]only depending on the size of the ambient space and the entropic regularization magnitude.

*Proof.* u Lemma D.4 shows that $\pi_\infty^\star$ is a solution of an optimal transport problem with a continuous cost $\widehat{c}$. Moreover, $p_\infty^\star = \exp(\varphi_\infty^\star \oplus \psi_\infty^\star - \widehat{c})$ (see Theorem 4.2. [57]) and, from Lemma 4.11 in [57], the functions $\varphi_\infty^\star$ and $\psi_\infty^\star$ can be assumed to be continuous (since $\widehat{c}$ is). The result for $p_\infty^\star$ now follows from the compactness of $\mathcal{X}$ and $\mathcal{Y}$.

The result for $\bar{\pi}_n^\star$ follows from Lemma D.3 and the fact that $\bar{\pi}_n^\star$ is a probability, *i.e.*, $(\bar{\pi}_n^\star)_{ij} \leq 1$. $\quad\square$

**Lemma D.7.** If $\mathcal{X}$ and $\mathcal{Y}$ are compact, then

$$|Z_n - \widehat{Z}_n| \leq \mathcal{O}(\log(n))\|\widehat{\pi}_n^\star - \bar{\pi}_n^\star\|_1.$$

*Proof.* Denote, for any $i,j \in [n]^2$, $V_{i,j} := ((x_i, s_i), (y_j, w_j))$

$$|Z_n - \widehat{Z}_n| = |\langle c\,\bar{\pi}_n^\star - \hat{\pi}_n^\star\rangle + \varepsilon\left(\mathbf{KL}(\bar{\pi}_n^\star||\mu_n \otimes \eta_n) - \mathbf{KL}(\hat{\pi}_n^\star||\mu_n \otimes \eta_n)\right) + \lambda\left(\mathcal{L}_\mathbf{F}(\bar{\pi}_n^\star) - \mathcal{L}_\mathbf{F}(\widehat{\pi}_n^\star)\right)|$$
$$\leq \|c\|_\infty\|\bar{\pi}_n^\star - \hat{\pi}_n^\star\|_1 + \varepsilon\,|\mathbf{KL}(\bar{\pi}_n^\star||\mu_n \otimes \eta_n) - \mathbf{KL}(\hat{\pi}_n^\star||\mu_n \otimes \eta_n)| + \lambda\,|\mathcal{L}_\mathbf{F}(\bar{\pi}_n^\star) - \mathcal{L}_\mathbf{F}(\widehat{\pi}_n^\star)|.$$

We bound the two remaining terms as follows. First, by Lemma D.6, there exists $r, s > 0$ such that, $\mu \otimes \eta$-a.s., $\frac{1}{r} \leq p_\infty^\star(V_{i,j}) \leq r$ and $\frac{1}{s} \leq \frac{d\bar{\pi}_n^\star}{d(\mu_n \otimes \eta_n)} \leq n^2$.

*(i)*

$$\mathcal{L}_\mathbf{F}(\hat{\pi}_n^\star) - \mathcal{L}_\mathbf{F}(\bar{\pi}_n^\star) \geq \Big\langle 2\sum_{s,w\in\{0,1\}^2} \langle \xi_{sw}, \bar{\pi}_n^\star\rangle \xi_{sw}, \hat{\pi}_n^\star - \bar{\pi}_n^\star\Big\rangle,$$

and

$$\mathcal{L}_\mathbf{F}(\bar{\pi}_n^\star) - \mathcal{L}_\mathbf{F}(\hat{\pi}_n^\star) \geq \Big\langle 2\sum_{s,w\in\{0,1\}^2} \langle \xi_{sw}, \hat{\pi}_n^\star\rangle \xi_{sw}, \bar{\pi}_n^\star - \hat{\pi}_n^\star\Big\rangle,$$

hence, for $n$ sufficiently large,

$$|\mathcal{L}_\mathbf{F}(\bar{\pi}_n^\star) - \mathcal{L}_\mathbf{F}(\hat{\pi}_n^\star)| \leq 2\max\{\|\sum_{s,w\in\{0,1\}^2} \langle \xi_{sw}, \hat{\pi}_n^\star\rangle \xi_{sw}\|_\infty, \|\sum_{s,w\in\{0,1\}^2} \langle \xi_{sw}, \bar{\pi}_n^\star\rangle \xi_{sw}\|_\infty\}\|\bar{\pi}_n^\star - \hat{\pi}_n^\star\|_1$$
$$\leq 8\max\{\|\hat{\pi}_n^\star\|_\infty, \|\bar{\pi}_n^\star\|_\infty\}\|\bar{\pi}_n^\star - \hat{\pi}_n^\star\|_1$$
$$\leq \mathcal{O}(1)\|\bar{\pi}_n^\star - \hat{\pi}_n^\star\|_1,$$

where we used the fact that $\xi_{sw} \in [-1, 1], \forall s, w \in \{0, 1\}$.

*(ii)* Similarly as in Lemma D.4, we can prove

$$\mathbf{KL}(\hat{\pi}_n^\star||\mu_n \otimes \eta_n) - \mathbf{KL}(\bar{\pi}_n^\star||\mu_n \otimes \eta_n) \geq \Big\langle \log\left(\frac{d\bar{\pi}_n^\star}{d(\mu_n \otimes \eta_n)}\right), \hat{\pi}_n^\star - \bar{\pi}_n^\star\Big\rangle,$$

and

$$\mathbf{KL}(\bar{\pi}_n^\star||\mu_n \otimes \eta_n) - \mathbf{KL}(\hat{\pi}_n^\star||\mu_n \otimes \eta_n) \geq \Big\langle \log\left(\frac{d\hat{\pi}_n^\star}{d(\mu_n \otimes \eta_n)}\right), \bar{\pi}_n^\star - \hat{\pi}_n^\star\Big\rangle,$$

hence, for $n$ sufficiently large,

$$|\mathbf{KL}(\bar{\pi}_n^\star||\mu_n \otimes \eta_n) - \mathbf{KL}(\hat{\pi}_n^\star||\mu_n \otimes \eta_n)|$$
$$\leq \max\{\|\log\left(\frac{d\hat{\pi}_n^\star}{d(\mu_n \otimes \eta_n)}\right)\|_\infty, \|\log\left(\frac{d\bar{\pi}_n^\star}{d(\mu_n \otimes \eta_n)}\right)\|_\infty\}\|\bar{\pi}_n^\star - \hat{\pi}_n^\star\|_1 \leq \mathcal{O}(\log(n))\|\bar{\pi}_n^\star - \hat{\pi}_n^\star\|_1.$$

$\square$

**Lemma D.8.** Let $\widehat{\pi}_n^\star$ be an $n \times n$ matrix with $ij$ entry defined by $(\widehat{\pi}_n^\star)_{ij} = p_\infty^\star(x_i, y_j)/n^2$ where $p_\infty^\star$ is a continuous function satisfying $\langle p_\infty^\star(x, \cdot), \eta\rangle = \langle p_\infty^\star(\cdot, y), \mu\rangle = 1$[4]. Then

$$\mathbb{E}\|\widehat{\pi}_n^\star \mathbf{1} - 1/n\mathbf{1}\|_1 \leq \mathcal{O}(\sqrt{n}^{-1}) \quad \text{and} \quad \mathbb{E}\|(\widehat{\pi}_n^\star)^T \mathbf{1} - 1/n\mathbf{1}\|_1 \leq \mathcal{O}(\sqrt{n}^{-1}).$$

---
[4]This condition is a consequence of $\widehat{\pi}_\infty^\star \in \Pi(\mu, \eta)$.

*Proof.* We consider only the bound of $\|\widehat{\pi}_n^\star \mathbf{1} - 1/n\mathbf{1}\|_1$ since the other is analogous. Note that

$$\|\widehat{\pi}_n^\star \mathbf{1} - 1/n\mathbf{1}\|_1 = \sum_i |1/n - 1/n^2 \sum_j (\hat{\pi}_n^\star)_{ij}| = 1/n \sum_i |1 - 1/n \sum_j p_\infty^\star(x_i, y_j)|$$

$$\leq \frac{\sqrt{n}}{n} \sqrt{\sum_i \left(1 - 1/n \sum_j p_\infty^\star(x_i, y_j)\right)^2} = \sqrt{\frac{1}{n} \sum_i \left(1 - 1/n \sum_j p_\infty^\star(x_i, y_j)\right)^2}$$

where the inequality follows from Cauchy-Schwartz. To finish note that the proof of Lemma 14 in [65] only relies on the marginal constraints $\langle p_\infty^\star(x, \cdot), \eta \rangle = \langle p_\infty^\star(\cdot, y), \mu \rangle = 1$ and the boundedness of $p_\infty^\star$. Consequently,

$$\mathbb{E}\frac{1}{n} \sum_i \left(1 - 1/n \sum_j p_\infty^\star(x_i, y_j)\right)^2 \leq \mathcal{O}(n^{-1})$$

and the result then follows by Jensen's inequality. $\qquad\square$

# E   Experimental details and results for cost learning

## E.1   Experimental details

Data $X$ and $Y$ are generated from mixtures of two bivariate Gaussian components, the sensitive attribute serving as an indicator of the Gaussian component from which each sample originates. In all the experiments, the strength of the entropic regularization is fixed at $\epsilon = 1$ and only fairness regularization varies. The base cost used is the squared euclidean distance.

For the penalized approach, the problem is solved by using the implementation of the Generalized Conditional Gradient algorithm provided by the Python package Python Optimal Transport (POT) [28, 29].

For the cost learning approach, the problem is solved using Adam algorithm with a learning rate fixed at $5 \times 10^{-3}$. The inner problem is solved using the POT implementation of the Sinkhorn algorithm. The gradient of the bilevel objective is computed by backpropagation in the Sinkhorn iterations.

## E.2   Results for the cost learning approach

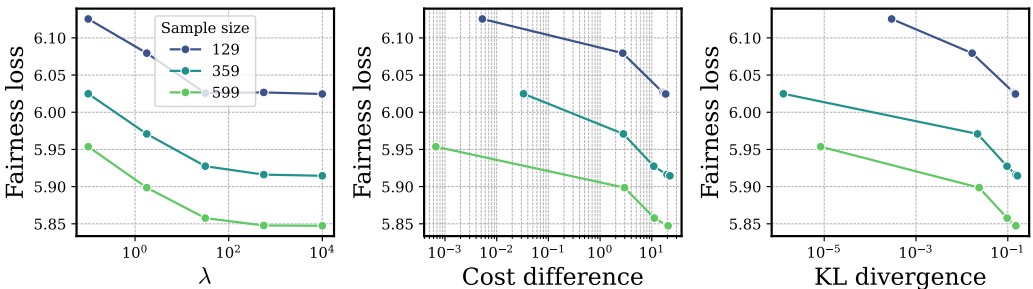

Figure 3: Fairness vs. penalty (**left**), and fairness-KL divergence (**center**) and fairness-cost difference (**right**) trade-offs (w.r.t. the non penalized problem) for varying sample sizes for the **cost-learning algorithm** with cost-sensitive fairness loss.

