# OpenReview forum: "Optimal Transport under Group Fairness Constraints"
_EurIPS.cc/2025/Workshop/UPLB — UPLB2025_

### Official Review · Reviewer_HTtf · 2025-10-26
**The main text is well-crafted and relevant to the workshop, so it is recommended for acceptance.**

**Rating:** 7
**Confidence:** 3

**Review:**

## Brief Summary

The paper examined the theoretical aspects of group matching under entropic optimal transport, subject to externally imposed fairness constraints. Because meeting the imposed fairness constraints could result in unaffordable costs, the paper proposed two methods for relaxing the constraints. Specifically, the paper proposed transforming the initial optimal transport problem into penalized correspondence, also referred to as **penalized optimal transport**. Alternatively, it considered a **cost learning** approach consisting of a bi-level optimization problem with a family of alternative, parameterized cost functions close to a baseline function. This parametrization can be performed using neural networks. The paper's main contributions include formulating the problem, presenting adapted algorithms, and deriving the sample complexity analysis for penalized optimal transport. Regarding the numerical experiments, the paper presented simulations of a Gaussian mixture that demonstrated a lower fairness loss corresponds to a higher bias with respect to an unfair optimal transport plan.

## Suggestions and Conclusions

The main body of the paper is clear and well-structured, and contains nice, illustrative examples. I regrettably did not have the chance to check every line in the appendix proof, but it appears to be solid. The proof sketch in the main text is also helpful. Moreover, although it may slightly diverge from the listed main goals and themss of the workshop, I still feel it is relevant and can foster meaningful discussion from the perspective of optimal transport. **Therefore, I recommend an acceptance.**

I would also like to list a few minor points here:
1. I probably misunderstood, or there may be a typo. I thought it was over $\mathcal S$ and $\mathcal W$, respectively, on line 46.
2. The comparison in Appendix C may not be thorough enough. Some toy examples and/or simulations in addition to Section 4 would be greatly appreciated. Moreover, due to the page limit, Section 3.2, the part about the bi-level formulation, seems too concise. For instance, it might be worth mentioning what kind of discrepancy measure can be employed and what the consequences would be, so readers can get a better sense of its generality.
3. It seems also due to the page limit, the numerical illustration section (section 4) in the main text is not self-contained, making it difficult to understand at first. It would be nice to reconstruct this section. Or alternatively, it would be greatly appreciated if you could mention at the beginning of this section that the setup details are presented in Appendix E, as well as provide more details in the appendix itself.
	1. There's a redundant "function" word on line 160.
	2. In general, any discussion of the numerical results would be greatly appreciated because the paper lacks discussion even in the appendix. For instance, for small sample sizes, there always seems to be a plateau in the penalized case. Could you explain why that is the case? Could you explain that as well? Could you also explain why the middle figure suddenly drops for the medium sample size?
	3. It would be helpful to distinguish the $\lambda$ between the two strategies using different symbols for each, e.g., $\lambda_p$ and $\lambda_b$.
	4. It seems that the neural network used for Appendix E is unspecified.

---

### Decision · Program_Chairs · 2025-11-03

Accept (Poster)